# Discovery of the 1-naphthylamine biodegradation pathway reveals a broad-substrate-spectrum enzyme catalyzing 1-naphthylamine glutamylation

Shu-Ting Zhang[1†], Shi-Kai Deng[1†], Tao Li[1], Megan E Maloney[2], De-Feng Li[3], Jim C Spain[2]*, Ning-Yi Zhou[1]*

[1]State Key Laboratory of Microbial Metabolism, Joint International Research Laboratory of Metabolic and Developmental Sciences, and School of Life Sciences and Biotechnology, Shanghai Jiao Tong University, Shanghai, China; [2]Center for Environmental Diagnostics and Bioremediation, University of West Florida, Pensacola, United States; [3]State Key Laboratory of Microbial Resources, Institute of Microbiology, Chinese Academy of Sciences, Beijing, China

**Abstract** 1-Naphthylamine (1NA), which is harmful to human and aquatic animals, has been used widely in the manufacturing of dyes, pesticides, and rubber antioxidants. Nevertheless, little is known about its environmental behavior and no bacteria have been reported to use it as the growth substrate. Herein, we describe a pathway for 1NA degradation in the isolate *Pseudomonas* sp. strain JS3066, determine the structure and mechanism of the enzyme NpaA1 that catalyzes the initial reaction, and reveal how the pathway evolved. From genetic and enzymatic analysis, a five gene-cluster encoding a dioxygenase system was determined to be responsible for the initial steps in 1NA degradation through glutamylation of 1NA. The γ-glutamylated 1NA was subsequently oxidized to 1,2-dihydroxynaphthalene which was further degraded by the well-established pathway of naphthalene degradation via catechol. A glutamine synthetase-like (GS-like) enzyme (NpaA1) initiates 1NA glutamylation, and this enzyme exhibits a broad substrate selectivity toward a variety of anilines and naphthylamine derivatives. Structural analysis revealed that the aromatic residues in the 1NA entry tunnel and the V201 site in the large substrate-binding pocket significantly influence NpaA1's substrate preferences. The findings enhance understanding of degrading polycyclic aromatic amines, and will also enable the application of bioremediation at naphthylamine contaminated sites.

## Editor's evaluation

This important work identifies a *P. aeruginosa* strain and enzyme that can degrade 1-naphthylamine, a harmful industrial pollutant. Data resulting from in vivo, biochemical, and structural approaches are compelling. This paper would be of major interest to biologists and enzymologists studying biodegradation of industrial pollutants.

## Introduction

Aromatic amines include monocyclic aromatic amines, such as aniline and chlorinated derivatives, and polycyclic aromatic amines, such as naphthylamine and its derivatives. Aromatic amines have been widely utilized as raw materials for manufacturing dyes, pharmaceuticals, and agrochemicals (*Palmiotto et al., 2001*). Anthropogenic activities have led to their widespread release into the environment

*For correspondence:
jspain@uwf.edu (JCS);
ningyi.zhou@sjtu.edu.cn (NYZ)

†These authors contributed equally to this work

**Competing interest:** The authors declare that no competing interests exist.

(*Akyüz and Ata, 2006*; *Dupret et al., 2011*). The United States produced 1050 thousand metric tons of aniline in 2013 (*Council, 2023*). Meanwhile, China's export volume of 1-naphthylamine, 2-naphthylamine, and their derivatives amounted to 19.8 thousand metric tons in 2013 (*Network, 2018*). Several aromatic amines are potentially harmful to human health, with both aniline and naphthylamines increasing the risk of bladder tumors (*Chung, 2000*; *Ferraz et al., 2012*). These compounds are often found in mixtures in the environment as impurities or by-products (*IARC, 1974*). Although various aromatic amines cause environmental pollution and threaten human health, their transport and fate are poorly understood.

Currently, only a few anilines and their derivatives have been reported to be degraded by microorganisms (*Król et al., 2012*; *Lee et al., 2008*; *Qu and Spain, 2011*; *Takeo et al., 2013*). Under aerobic conditions, microbial degradation plays a major role in the elimination of aniline, and the molecular basis has been well-established in several bacteria (*Fukumori and Saint, 1997*; *Król et al., 2012*; *Liang et al., 2005*; *Takeo et al., 2013*). The aniline dioxygenase (AD) enzyme system is responsible for converting aniline to catechol, which is then assimilated via the widespread *meta/ortho*-cleavage pathways. AD comprises a γ-glutamylanilide synthase, a glutamine amidotransferase (GAT)-like enzyme, and a two-component Rieske-type aromatic compound dioxygenase (*Takeo et al., 2013*). The γ-glutamylanilide synthase catalyzes ATP-dependent ligation of L-glutamate to aniline to generate γ-glutamylanilide (γ-GA). Due to the cytotoxicity of high concentration of γ-GA, the GAT-like enzyme can hydrolyze γ-GA to aniline so as to maintain γ-GA at a proper level inside the cell. Subsequently, the dioxygenase catalyzes the transformation of γ-GA to catechol (*Takeo et al., 2013*). Previous studies established that the γ-glutamylanilide synthetase and oxygenase of AD play dominant roles in the substrate specificity of the pathway (*Ang et al., 2009*; *Ji et al., 2019*). To enhance the engineering potential of AD systems, the structure and mechanism of the oxygenase have been revealed to broaden its substrate range (*Ang et al., 2007*). However, limited research has been conducted on the γ-glutamylanilide synthetase.

In contrast to the well-established degradation of anilines, little is known about the environmental fate and biodegradation of 1NA, and bacteria able to assimilate it have not been reported. Here, we report the isolation of an aerobic bacterial strain able to use both 1NA and aniline as growth substrates through selective enrichment with samples from a 1NA-manufacturing site. Detailed investigations established the degradation pathway of 1NA and the genes encoding the enzymes involved. An enzyme system homologous to the AD system mentioned above catalyzes glutamylation of 1NA and subsequent oxidation of the product to dihydroxynaphthalene. Previous research has suggested that the substrate specificity of the glutamylating enzyme, which is responsible for the initial glutamylation of aromatic amines, plays a crucial role in determining the range of substrates that can be degraded (*Ang et al., 2007*; *Ji et al., 2019*). Therefore, we elucidated the detailed structure and mechanism underlying its substrate specificity. The results will enable us to predict and enhance 1NA biodegradation at contaminated sites and provide the basis for a better understanding of the degradation of other polycyclic aromatic amines.

## Results

### *Pseudomonas* sp. strain JS3066 is a 1NA degrader

Selective enrichment with 1NA as the growth substrate yielded an isolate that grew aerobically on 1NA as the sole carbon and nitrogen source (*Figure 1A*). It could also utilize aniline for growth (data not shown). A BLASTN search against the sequences on the National Center for Biotechnology Information (NCBI) website (http://www.ncbi.nlm.nih.gov/) revealed that its 16S rRNA gene sequence shows 99.93% and 99.74% identity with *Pseudomonas* sp. DY-1 (GenBank accession number: CP032616.1) and *Pseudomonas* sp. TCU-HL1 (GenBank accession number: CP015992.1), respectively. Thus, the 1NA degrader was identified as *Pseudomonas* sp. strain JS3066.

### A proposed 1NA conversion cluster locates on a plasmid

The genome of strain JS3066 comprises two replicons, one circular chromosome (6,093,500 bp, G+C content of 62.94%), and one circular plasmid designated pJS3066 (109,408 bp, G+C content of 63.15%). The replication initiator protein TrfA of plasmid pJS3066 shares 100% amino acid sequence identity with that of plasmid pTP6, which indicates that pJS3066 likely belongs to the IncP-1β subgroup

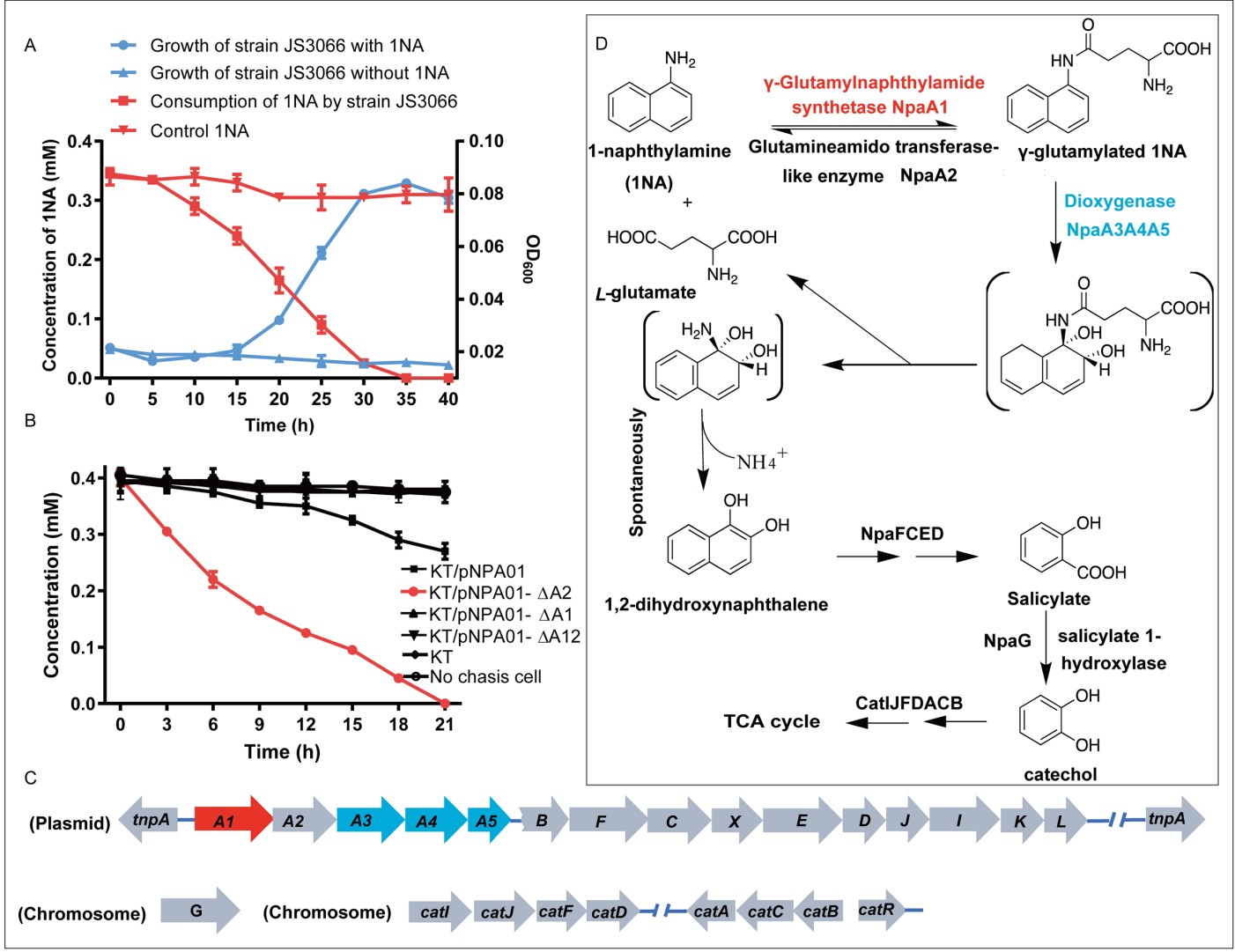

**Figure 1.** Degradation of 1-Naphthylamine (1NA) by *Pseudomonas* sp. strain JS3066. (**A**) Growth with 1NA as the sole carbon and nitrogen source. (**B**) 1NA conversion by cell suspensions of *P. putida* KT2440-ΔcatAΔggt harboring pNPA01 (*npaA1* to *npaA5*), pNPA01-ΔA1 (*npaA2* to *npaA5*), pNPA01-ΔA2 (*npaA1* and *npaA3* to *npaA5*), and pNPA01-ΔA12 (*npaA3* to *npaA5*). The ability of *P. putida* KT2440-ΔcatAΔggt to convert 1NA together with the spontaneous decomposition of 1NA were also determined as control. The error bars represent the standard deviation while the points represent the mean of two independent measurements. (**C**) Organization of the genes involved in 1NA degradation by strain JS3066. Gene *tnpA* encodes a transposase. Proposed functions of the rest of genes are presented in *Supplementary file 1*. (**D**) Proposed pathway of 1NA catabolism in *Pseudomonas* sp. strain JS3066. TCA, tricarboxylic acid. Unstable compounds are enclosed in brackets.

The online version of this article includes the following source data and figure supplement(s) for figure 1:

**Source data 1.** Raw data for the *Figure 1A* and *Figure 1B*.

**Figure supplement 1.** Gas chromatography-mass spectrometry (GC-MS) analysis of the intermediate accumulated during 1-Naphthylamine (1NA) transformation by the cell suspension of *P. putida* KT2440-ΔcatAΔggt harboring pNPA01.

**Figure supplement 2.** Conversion of aniline by strain *P. putida* KT2440-ΔcatAΔggt harboring plasmid pNPA01 (recombinant KT/pNPA01).

**Figure supplement 3.** Conversion of γ-glutamylated 1-naphthylamine by recombinant strain KT/pNPA01-ΔA12.

**Figure supplement 4.** High-performance liquid chromatography (HPLC) chromatographs of salicylate standard, catechol standard, and the reaction product of salicylate conversion catalyzed by the recombinant protein NpaG.

of plasmids (*Stenger and Lee, 2011*; *Thorsted et al., 1996*). Collectively, the whole genome of strain JS3066 is 6,202,908 bp with an average G+C content of 62.95%.

The ability of the isolate to degrade aniline coupled with the fact that 1NA is an analog of aniline prompted the hypothesis that the initial reactions of 1NA and aniline catabolism might be

mediated by the similar genetic determinants within strain JS3066. Therefore, the well-studied aniline dioxygenase-encoding gene set (*atdA1A2A3A4A5*; accession number: D86080.1) from *Acinetobacter* sp. YAA was used as the query to search the genome of strain JS3066. A cluster of genes closely related to those encoding aniline dioxygenase is located on plasmid pJS3066. The putative naphthylamine dioxygenase-encoding genes, designated *npa* (1-<u>n</u>a<u>p</u>hthyl<u>a</u>mine), are encoded in the order *npaA1A2A3A4A5* (*Figure 1C*). BLASTp analyses of the deduced amino acid sequences of NpaA1A2A3A4A5 against the NCBI database revealed high identities with the γ-glutamylanilide synthase (AtdA1), the GAT-like enzyme (AtdA2), the two-component Rieske-type aromatic compound dioxygenase (AtdA3A4), and the reductase component (AtdA5) in *Acinetobacter* sp. YAA (*Supplementary file 1*). No other potential aniline dioxygenase homologs were detected in the genome.

## NpaA1A3A4A5 convert 1NA to 1,2-dihydroxynaphthylene

To investigate the roles of *npaA1A2A3A4A5* genes in the initial oxidation of 1NA, plasmids carrying different combinations of the five genes were constructed and introduced into *P. putida* KT2440-Δ*catA*Δ*ggt* which is unable to catabolize catechol. Then the abilities of recombinant KT/pNPA01 (including *npaA1A2A3A4A5* genes) and its derivatives to metabolize 1NA were analyzed. Recombinant KT/pNPA01 and KT/pNPA01-ΔA2 (including *npaA1*, *npaA3* to *npaA5*) were able to transform 1NA (*Figure 1B*). However, cells harboring either *npaA2A3A4A5* genes (recombinant KT/pNPA01-ΔA1) or *npaA3A4A5* genes (KT/pNPA01-ΔA12) were unable to transform 1NA. This indicates that NpaA1 is an essential component in the conversion of 1NA, while NpaA2 is not necessary. According to sequence alignment, NpaA2 is a GAT-like protein, sharing 98% sequence identity with AtdA2 (*Takeo et al., 2013*). NpaA2 is likely similar to AtdA2, hydrolyzing γ-glutamylated 1NA into 1NA, thus reversing the γ-glutamylated 1NA formation catalyzed by NpaA1. Experiments on whole-cell biotransformation indicate that the strain *P. putida* KT2440-Δ*catA*Δ*ggt*, which includes NpaA1A3A4A5 but lacks NpaA2, can still convert 1NA into 1,2-dihydroxynaphthalene (*Figure 1B*). Gas chromatography-mass spectrometry (GC-MS) analysis revealed that the main product formed during the transformation of 1NA by recombinant KT/pNPA01 was 1,2-dihydroxynaphthalene (*Figure 1—figure supplement 1*). Notably, recombinant KT/pNPA01 retained the ability to transform aniline into catechol (*Figure 1—figure supplement 2*).

To enable further characterization, NpaA1 was expressed and purified as an N-terminal Strep II-tagged fusion protein. 1NA conversion by NpaA1 was investigated in a reaction mixture similar to that used previously for the transformation of aniline to γ- glutamylated aniline (*Ji et al., 2019*). 1NA ($\lambda_{max}$, 310 nm) was transformed by purified NpaA1 to a new product with maximum absorption at around 287 nm (*Figure 2—figure supplement 1*). The HPLC retention time of the product was equal to that of the authentic γ-glutamylated 1NA and ultra-performance liquid chromatography-quadrupole-time of flight-mass spectrometry (UPLC-QTOF MS) analysis showed that the product had a molecular ion at *m/z* 273.1231 [M+H]⁺, which is identical to that of authentic γ-glutamylated 1NA (*Figure 2C and D*). During γ-glutamylation of 1NA, the 1NA consumption (0.16 mmol) was almost equivalent to the total accumulation of γ-glutamylated 1NA (0.157 mmol) (*Figure 2A and B*). In view of the above analyses, NpaA1 was established as a γ-glutamylnaphthylamide synthase catalyzing ligation of 1NA and L-glutamate to form γ-glutamylated 1NA.

Although recombinant KT/pNPA01-ΔA12 failed to transform 1NA, it was able to catalyze the conversion of γ-glutamylated 1NA into 1,2-dihydroxynaphthalene, indicating that γ-glutamylated 1NA is an intermediate and a direct substrate for the dioxygenase NpaA3A4A5 (*Figure 1—figure supplement 3*). Experiments with different combinations of *npaA1A2A3A4A5* genes have established that only four Npa proteins, namely, NpaA1 (GS-like enzyme), NpaA3 (large subunit of oxygenase component of dioxygenase), NpaA4 (small subunit of oxygenase component of dioxygenase), and NpaA5 (reductase component of dioxygenase) are essential for the conversion of 1NA to 1,2-dihydroxynaphthylene (*Figure 1B and D*).

## Proposed pathway for 1,2-dihydroxynaphthalene degradation in strain JS3066

Downstream of the *npaA1A2A3A4A5* genes lie *npaBFCXEDJIKL* genes (*Figure 1C* and *Supplementary file 1*), whose products, except for *napX* encoding a transposase, are closely related to those involved in naphthalene degradation by *Ralstonia* sp. strain U2 and related strains (*Fuenmayor*

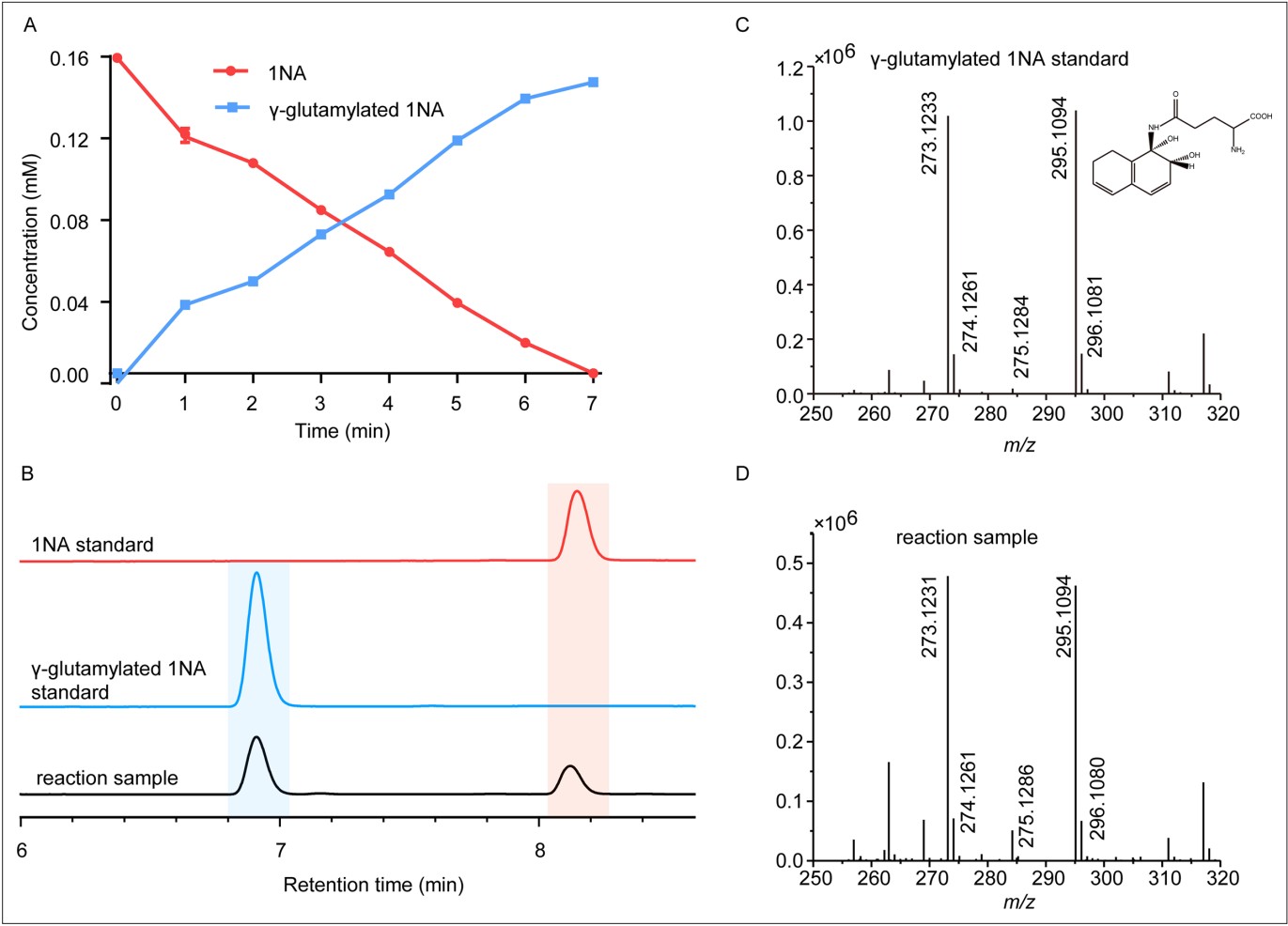

**Figure 2.** Conversion of 1-Naphthylamine (1NA) by recombinant protein NpaA1. (**A**) The time course of 1NA degradation and γ-glutamylated 1NA accumulation. The error bars represent the standard deviation while the points represent the mean of two independent measurements. (**B**) High-performance liquid chromatography (HPLC) profiles of 1NA standard, γ-glutamylated 1NA standard, and the reaction products of 1NA conversion catalyzed by NpaA1. The detection wavelength was 280 nm. (**C–D**) UPLC/QTOF-MS analysis of the intermediate captured during 1NA transformation by NpaA1. The mass spectra of authentic γ-glutamylated 1NA (**C**) and reaction product γ-glutamylated 1NA (**D**).

The online version of this article includes the following source data and figure supplement(s) for figure 2:

**Source data 1.** Raw data of *Figure 2A, C and D*.

**Figure supplement 1.** Spectral changes during the transformation of 1-Naphthylamine (1NA) by purified N-terminal Strep II-tagged NpaA1.

**Figure supplement 1—source data 1.** Raw data for the spectral changes shown in *Figure 2—figure supplement 1*.

*et al., 1998*; *Zhou et al., 2001*). In strain U2, a putative aldolase-encoding gene (*nagQ*) lies between the 1,2-dihydroxynaphthalene dioxygenase-encoding gene (*nagC*) and the *trans-o*-hydroxybenzylidenepyruvate hydratase-aldolase-encoding gene (*nagE*); in strain JS3066, however, *npaX*, a putative transposase-encoding gene, lies there instead.

Under oxic conditions, naphthalene is often first oxidized to salicylate, which is further channeled to tricarboxylic acid (TCA) cycle intermediates either via gentisate or catechol catabolic pathways. The metabolism of naphthalene via catechol has been studied in *P. putida* G7 (bearing the catabolic plasmid NAH7) (*Sota et al., 2006*) and in *P. putida* NCIB 9816–4 (bearing the catabolic plasmid pDTG1) (*Dennis and Zylstra, 2004*) at the genetic level. Likewise, the gentisate pathway has been found in *Ralstonia* sp. strain U2 (*Fuenmayor et al., 1998*; *Zhou et al., 2001*) and *Polaromonas naphthalenivorans* CJ2 (*Jeon et al., 2006*). Based on the above bioinformatic analysis and analogy with the established pathways for naphthalene degradation, the putative genes sufficient to encode the lower degradation pathway of 1NA would be complete, except for the absence of a gene responsible

for salicylate conversion. Given the fact the gentisate-catabolic genes are intact in strain JS3066 (*Figure 1C and D*), we searched for a gene which would catalyze the conversion of salicylate to gentisate. Contrary to expectation, no putative salicylate 5-hydroxylase-encoding gene was found either upstream or downstream of npa*A1A2A3A4A5BFCXEDJIKL* genes. We, therefore, searched the whole genome for genes encoding putative enzymes capable of catalyzing (ⅰ) transformation of salicylate to catechol (*Dennis and Zylstra, 2004*; *Jouanneau et al., 2007*; *Sota et al., 2006*), (ⅱ) conversion of salicylate to gentisate via salicylyl-CoA and gentisyl-CoA (*Zhou et al., 2021*), and (ⅲ) direct ring fission of salicylate to 2-oxohepta-3,5-dienedioic acid (*Hintner et al., 2001*). The search revealed a putative salicylate 1-hydroxylase-encoding gene, designated *npaG* and located on the chromosome of strain JS3066. The product of *npaG* exhibits 77% identity with NahG of *P. putida* G7 (*Sota et al., 2006*). NpaG was functionally expressed and found to catalyze the conversion of salicylate to catechol with a specific activity of $15.0 \pm 1.3$ U mg$^{-1}$ (*Figure 1—figure supplement 4*). The bioinformatic analysis did not reveal candidate genes encoding enzymes that catalyze any of the known alternative routes of salicylate metabolism. The above results supported the hypothesis that *npaBFCXEDJIKL-G* genes, products of which catalyze the conversion of 1,2-dihydroxynaphthalene to catechol, are involved in 1NA degradation in strain JS3066. The putative genes responsible for encoding the well-defined *ortho*-cleavage pathway of catechol degradation are located on the chromosome in strain JS3066 (*Figure 1C*). The genes show high similarities to their counterparts in other strains that degrade catechol (*Supplementary file 1*).

## NpaA1 converts multiple aromatic amine substrates to γ-glutamylated aromatic amines

Conversion to the corresponding γ-glutamylated amines is an essential step for aromatic amines oxidation, because the dioxygenase is unable to directly act on the amines. Furthermore, the glutamylating enzyme plays an essential role in the substrate specificity of the pathway (*Ji et al., 2019*). The activity of NpaA1 and AtdA1 against various aniline and naphthylamine derivatives have been determined in this study. Enzyme assays with purified NpaA1 revealed its activity not only against polycyclic aromatic amines such as 1NA and 2-naphthylamine, but also monocyclic aromatic amines and their chlorinated derivatives such as aniline and 3,4-dichloroaniline (*Table 1*). In contrast, AtdA1 from an aniline-degrading strain YAA as well as other reported glutamylating enzymes acting on chloroaniline only exhibited activity against monocyclic aniline derivatives (*Takeo et al., 2013*). It is worth noting that the in vitro activity of NpaA1 and AtdA1 heterologously expressed in *Escherichia coli* is significantly lower than their activity in their wild-type hosts. This issue was encountered in previous reports on AtdA1 (*Takeo et al., 2013*; *Ji et al., 2019*), though the reasons for the discrepancy remain unexplained. Here, the optimal pH of NpaA1 for enzymatic activity was 8.0 (*Figure 3—figure supplement 1A*), and the optimal temperature was 50 ° C (*Figure 3—figure supplement 1B*). The broad substrate specificity and high optimal temperature of NpaA1 indicate a potential for its biotechnological applications by protein engineering.

## NpaA1 is a GoaS (glutamyl organic amide synthetase) protein

Glutamine synthetase (GS), catalyzing the ATP-dependent synthesis of glutamine from glutamate and ammonium (*Harper et al., 2010*), is a member of an ancient and ubiquitous family of enzymes. GS enzymes can be divided into three distinct types, GSI, GSII, and GSIII (*Brown et al., 1994*; *de Carvalho Fernandes et al., 2022*). Similar to the nomenclature of GS based on functional characteristics, as various organic amine glutamine synthetases have been functionally identified, such as NpaA1, γ-glutamylanilide synthase, and γ-glutamylpolyamine synthetase (Gln3), we propose naming this enzyme class as GoaS. To investigate the relationships between GoaS and GS proteins, phylogenetic analysis of functionally identified GoaS and GS was conducted, revealing that NpaA1 is part of a well-separated branch containing 7 other GoaS proteins, except PA5508 and GlnA3 being located at a closely adjacent branch (*Figure 3A*). Among these GoaS proteins, besides the well-known AtdA1 for aniline degradation and newly identified NpaA1 for 1NA degradation, other enzymes are involved in isopropylamine degradation (IpuA) (*de Azevedo Wäsch et al., 2002*), chloroaniline degradation (DcaQ, TdnQ) (*Król et al., 2012*), putrescine utilization (PuuA) (*Kurihara et al., 2008*), etc. Although all members are involved in the degradation of organic amine compounds, none of them were reported

**Table 1.** Specific activities of NpaA1 and AtdA1 against different substrates.

| Substrate | NpaA1 Relative activity | AtdA1 Relative activity |
|---|---|---|
| (Aniline) | 100% | 100% |
| (1-naphthylamine) | 100% | 0 |
| (3-chloroaniline) | 76.5% | 29.4% |
| (3,4-dichloroaniline) | 31.4% | 0 |
| (2-naphthylamine) | 23.5% | 0 |
| (1,5-naphthalenediamine) | 5.9% | 0 |
| (1,8-naphthalenediamine) | 0 | 0 |
| (2,3-naphthalenediamine) | 7.8% | 0 |
| (2,7-naphthalenediamine) | 0 | 0 |

The specific activities of NpaA1 (25.5±0.7 U g$^{-1}$) and AtdA1 (8.5±2.1 U g$^{-1}$) against aniline were defined as 100%. The results are shown as averages ± standard deviations from two or more independent measurements.

The online version of this article includes the following source data for table 1:

**Source data 1.** Raw data for the specific activity of NpaA1 is shown in *Table 1*.

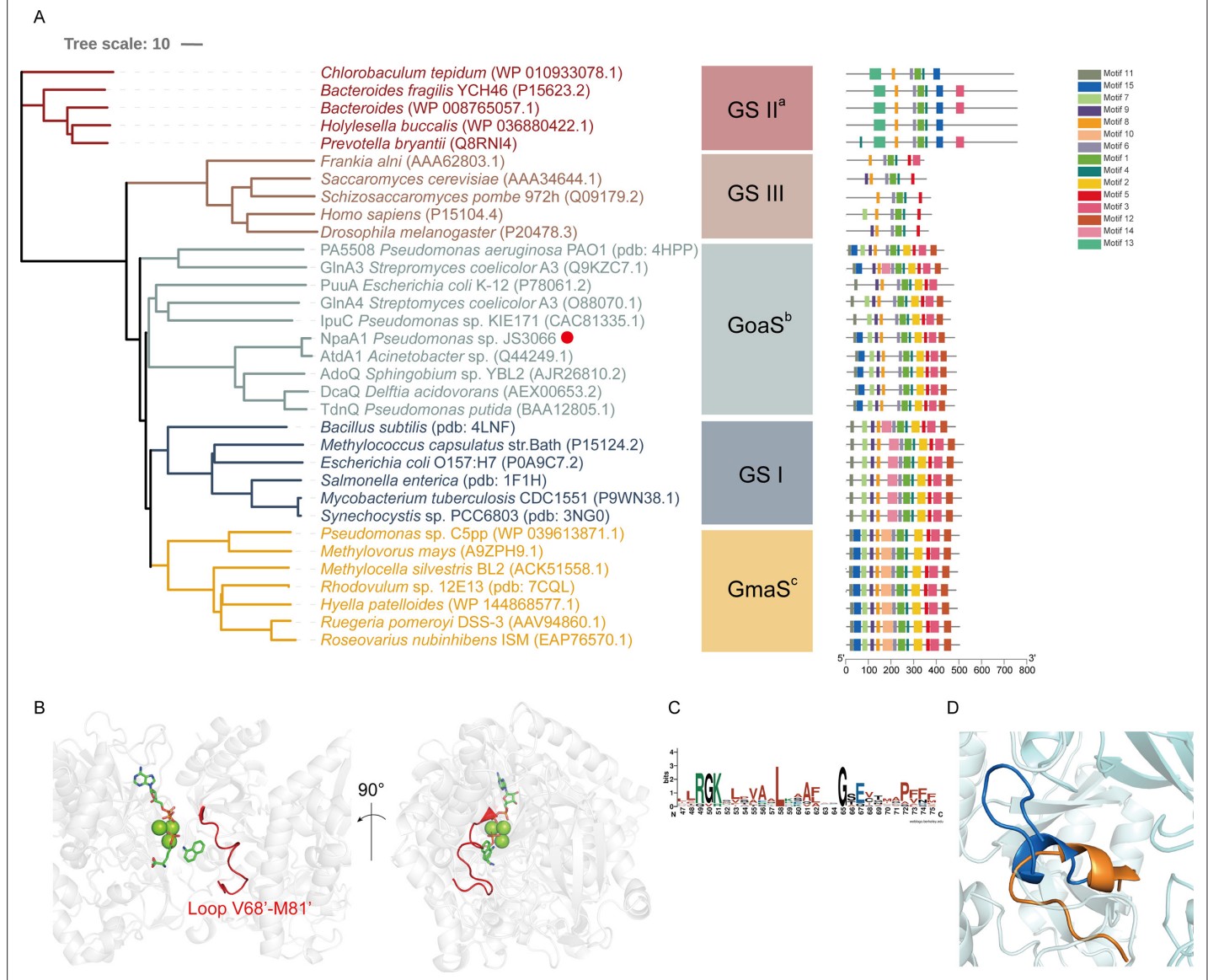

**Figure 3.** Phylogenetic analysis and motif analysis of GS-like enzymes. (**A**) Phylogenetic tree illustrating the evolutionary relationship between amino acid sequences of NpaA1 from *Pseudomonas* sp. strain JS3066 labeled with a red dot and other related sequences. (**B**) The loop V68'-M81' (represented as a red cartoon) in NpaA1–ADP–MetSox-P docking with 1NA complex. Ligands of this complex are colored in green. (**C**) Sequence logo of the 15th motif. (**D**) The overlapping structure of loop V68'-M81' of NpaA1 (in blue) and the corresponding region in *St*GS (in orange). GS: glutamine synthetase; GoaS: organic amine glutamine synthetases. GmaS: γ-Glutamylmethylamide synthetase.

The online version of this article includes the following source data and figure supplement(s) for figure 3:

**Figure supplement 1.** The optimal pH (**A**) and temperature (**B**) of wild-type NpaA1 activity.

**Figure supplement 1—source data 1.** Raw data for the NpaA1 activity shown in *Figure 3—figure supplement 1*.

**Figure supplement 2.** Motif order and spacing for glutamine synthetase (GS)-like proteins.

to catalyze ammonium conversion (*de Azevedo Wäsch et al., 2002*; *Ladner et al., 2012*; *Rexer et al., 2006*; *Takeo et al., 2013*).

GoaS enzymes including NpaA1 showed a closer relationship and more similar motif arrangement to GSI than other types of GS (*Figure 3A*). There are 11 conserved motifs between GSI and GoaS enzymes, but most members in GoaS have an additional 15th motif located in the N-terminal domain, which is absent in GSI (*Figure 3A* and *Figure 3—figure supplement 2*). In GS enzymes, the N-terminal domain contributes to the substrate binding of the enzyme (*Almassy et al., 1986*). The

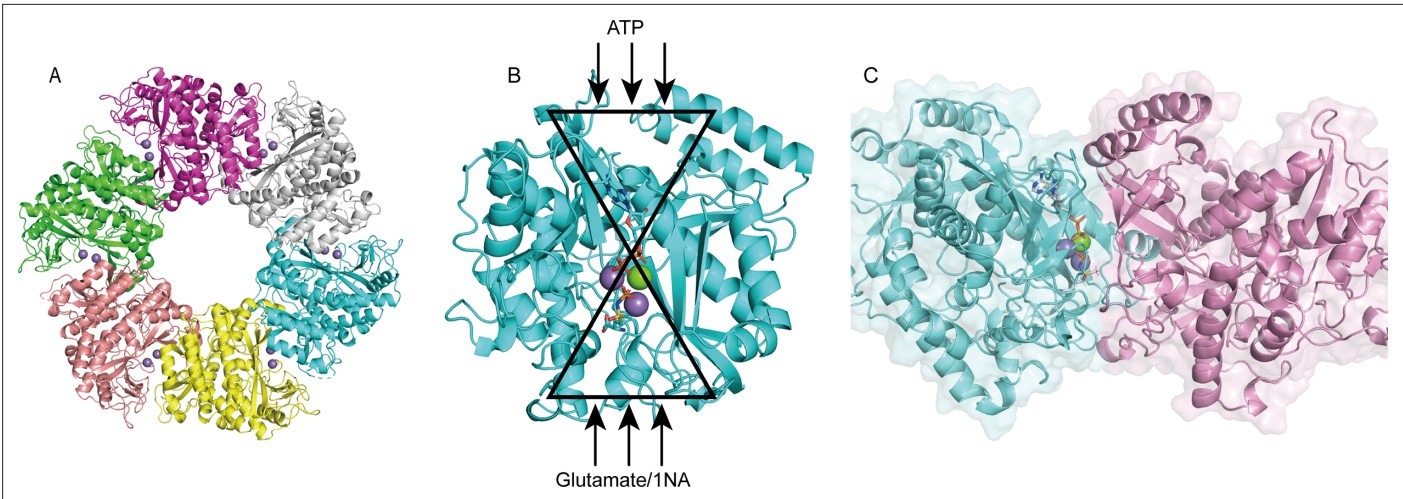

**Figure 4.** Structure analysis of NpaA1. (**A**) The overall structure of NpaA1. There are six monomers arranged as a hexamer in an asymmetric unit. The six monomers are in different colors. (**B**) The glutamine synthetase (GS) active site is illustrated as a bifunnel with the ATP, glutamate, and amine entry and binding sites at opposite ends. (**C**) Active site of NpaA1 located at the interface of adjacent subunits. The purple spheres represent the $Mn^{2+}$ ions, and the green spheres represent the $Mg^{2+}$ ions. Ligands are indicated by green sticks.

The online version of this article includes the following source data and figure supplement(s) for figure 4:

**Figure supplement 1.** Aggregate state analysis of NpaA1.

**Figure supplement 1—source data 1.** Raw data for Gle filtration shown in *Figure 4—figure supplement 1*.

**Figure supplement 1—source data 2.** Raw data for enzyme activities shown in *Figure 4—figure supplement 2*.

**Figure supplement 1—source data 3.** Raw data for the gel shown in *Figure 4—figure supplement 1*.

**Figure supplement 2.** Conserved residues and mutant analysis of ligand binding sites.

**Figure supplement 2—source data 1.** Raw data for enzyme assay of mutants shown in *Figure 4—figure supplement 2*.

presence of the 15th motif in NpaA1 suggests a distinct substrate binding mode for GoaS compared to GSI. This may allow NpaA1 to catalyze glutamylation reactions with a broad range of substrates.

## NpaA1 is a hexamer in solution

The crystal structure of apo-NpaA1 and two complexes with substrates or substrate analogs were obtained (*Supplementary file 2*). The model of NpaA1 built by Alphafold2 (*Jumper et al., 2021*) was used as the template for a molecular replacement for apo-NpaA1. Theoretical calculations and SDS-PAGE analysis showed that the molecular weight of the NpaA1 monomer was about 55 kDa. The results of gel filtration show two absorption peaks for purified NpaA1, indicating the presence of two distinct aggregation states in the solution. (*Figure 4—figure supplement 1A*). Multi-angle light scattering (MALS) analysis showed that NpaA1 exists in both monomeric and hexameric states in solution, with hexamers constituting approximately 36.7% of the total population (*Figure 4—figure supplement 1C*). It has been reported that glutamine synthetase also often exists in different oligomeric states in solution (*Joo et al., 2018*; *Travis et al., 2022*). The crystal structure of apo-NpaA1 revealed that there are six monomers arranged as a hexamer in an asymmetric unit (*Figure 4A*). The binding pocket is located at the interface between adjacent subunits (*Figure 4A and C*), and the bifunnel channel similar to GS protein refers to a structural feature characterized by two distinct channels that converge into the common active site. In apo NpaA1, there are two $Mn^{2+}$ ions binding to the active center, coordinated by glutamic acid and histidine residues (*Figure 4A* and *Figure 4—figure supplement 2A*). When the ATP analog AMPPNP binds to NpaA1, an additional $Mg^{2+}$ binds to the active site, facilitating the binding of the AMPPNP phosphate group (*Figure 4B*).

## NpaA1 has a large and hydrophobic active pocket

In order to elucidate the catalytic characteristics of NpaA1 towards organic amines, we conducted a comparative analysis of the differences in the substrate-binding pockets between NpaA1 and GSI family proteins. NpaA1 shows a similar architecture to GSI protein from *Salmonella typhimurium* (*St*GS)

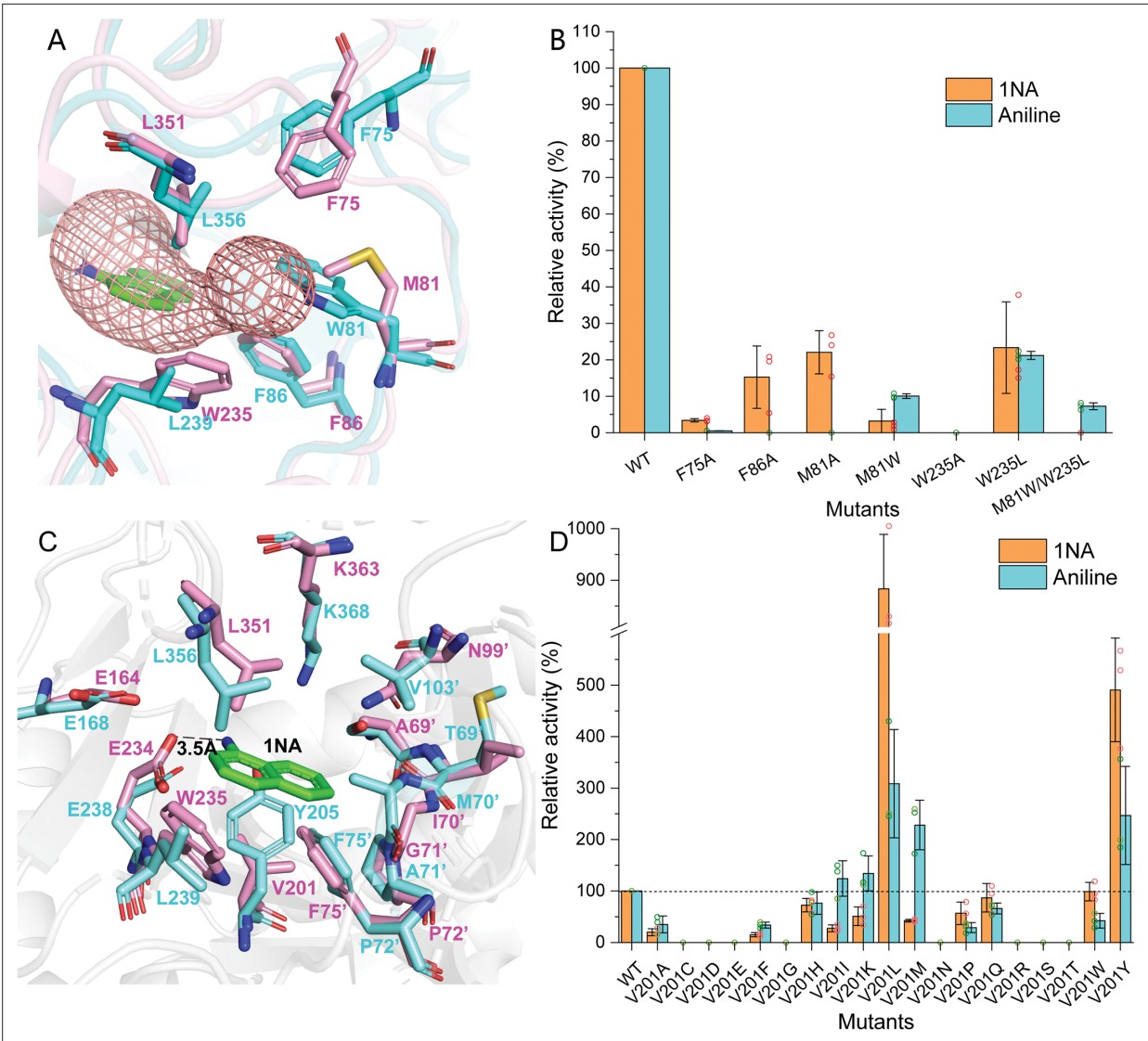

**Figure 5.** Residues involved in 1-Naphthylamine (1NA) binding. (**A**) The 1NA entry tunnel was predicted by Caver 3.0 (in red). The structures of NpaA1 (in pink) and AtdA1 (in cyan) overlap, with the amino acids contributing to the formation of tunnel bottlenecks highlighted as sticks. (**B**) Relative activity of mutants for 1NA entry tunnel. The activity of wild-type NpaA1 for 1NA is set to 100% (21.6±1.2 U/g), and wild-type NpaA1 showed equivalent activity to 1NA and aniline. Error bars represent the standard deviation of three biological replicates. (**C**) The aromatic amine binding pocket of NpaA1 is superimposed on AtdA1. 1NA is green, residues for NpaA1 are pink, and residues for AtdA1 are cyan. The black dashed line represents the hydrogen bond. (**D**) Relative activity of saturation mutants for V201. The activity of wild-type NpaA1 for 1NA is set to 100% (20.8±0.5 U/g), and wild-type NpaA1 showed equivalent activity to 1NA and aniline. Dots represent the relative activity of the respective biological replicates (n = 3) and error bars represent the standard deviation.

The online version of this article includes the following source data for figure 5:

**Source data 1.** Raw data for the relative activity of mutants is shown in **Figure 5**.

with a 1.2 Å root mean square deviation (RMSD) (**Gill and Eisenberg, 2001**), and γ-glutamylmethyl-amide synthetase from *Rhodovulum* sp. 12E13 (*Rh*GmaS) with a 1.0 Å RMSD for the aligned Cα coordinates (**Wang et al., 2021**). The coordinating residues involved in ATP, glutamate, and Mn$^{2+}$ binding are highly conserved between NpaA1 and GS (**Figure 4—figure supplement 2A and B**). The structural comparison between NpaA1 and *St*GS indicates a distinction in their ammonium binding sites. In contrast to the small negatively charged pocket for ammonium binding in GSI proteins, docking analysis of NpaA1 with 1NA suggested that 1NA binds at the interface of the adjacent monomer through electrostatic and hydrophobic interactions (**Figure 3B**). The positively charged amino group of 1NA

binds in a negatively charged pocket consisting of residues E232 and E234, and forms hydrogen bonds with residue E234 and the phosphate group of the intermediate γ-glutamyl phosphate (*Figure 5C*). The corresponding entry for 1NA in NpaA1 is surrounded by a long loop V68'-M81' which is located on part of the 15th motif in the N-terminal domain of the adjacent monomer (*Figure 3A–C*). The unique loop V68'-M81' seems to form part of the active pocket leading to a larger and more hydrophobic cavity (318.17 Å³) compared to that of the *St*GS protein (PDB ID:1F1H; 163.83 Å³) (*Figure 3D*). The enriched hydrophobic residues on the 15th motif of loop V68'-M81' (A69', I70', G71', P72', and F75') stabilize the aromatic ring of 1NA through hydrophobic interaction. The volume of the large pocket in NpaA1 is sufficient to accommodate 1NA, which also allows NpaA1 to accommodate a wider range of substrates. The 15th motif is present in the majority of GoaS enzymes responsible for catalyzing the glutamylation of organic amines. Similarly, GmaS, which possesses the 15th motif, also acts on organic amines as substrates (*Figure 3A*). The 15th motif is situated in the vicinity of the active pocket in GmaS (*Wang et al., 2021*) and GoaS, suggesting that this motif may play a role in facilitating the binding of organic compounds in these GS-like enzymes.

## The difference in the organic amine binding tunnels between NpaA1 and AtdA1

Despite the high identity (90%) between NpaA1 and AtdA1, only NpaA1 is capable of converting both monocyclic and polycyclic aromatic amines. To investigate how the differences in amino acids would affect substrate preferences and conversion rates, a predicted structure of AtdA1 was obtained from the AlphaFold2 protein structure database (*Jumper et al., 2021*). Structure overlapping of NpaA1 and AtdA1 suggested that the binding pockets in these two proteins are similar, while the variations in tunnel and binding pocket residues between NpaA1 and AtdA1 likely account for their different substrate specificities. We employed CAVER 3.0 software (*Chovancova et al., 2012*) to analyze potential tunnels for the entry of 1-naphthylamine or aniline into NpaA1 and AtdA1. The prediction of the tunnel in NpaA1 is based on structural docking after the removal of the ligand MetSox-P from the NpaA1—ADP—MetSox-P complex and subsequent docking of phosphorylated glutamate. 1NA docked in NpaA1 was selected as a mandatory site. The tunnel analysis in NpaA1 identified eight tunnels. Among the generated tunnels, a tunnel extending from the buried 1NA binding region to the surface of the bifunnel-shaped entry appears to be the most likely candidate for facilitating 1NA diffusion and overall catalysis in NpaA1 (*Figure 5A*). One of the predicted tunnels in AtdA1 overlaps with this tunnel in NpaA1, therefore, likely to be the amine binding channel.

In NpaA1, the indole ring of W235 is oriented parallel to the channel and likely engages in π-π interactions or cation-π interactions with substrates or surrounding aromatic amino acids, assisting in substrate entry. Structural analysis indicates that hydrophobic residues on the 15th motif contribute to the formation of the substrate tunnel (*Figure 5A*), with the most important ones being three hydrophobic aromatic amino acids (F75', F86', and W235) located at the tunnel bottleneck. Both NpaA1 and AtdA1 have three aromatic amino acids at the bottleneck, suggesting that these three aromatic amino acids may help in the proper positioning of aromatic amines in the tunnel. To identify the roles of these aromatic amine residues in regulating and controlling the substrate spectrum of NpaA1, a series of protein variants was generated with mutations in the NpaA1 tunnel, and the resultant mutant proteins were tested by enzyme assays. The enzyme assays of the NpaA1 mutants showed that F86A and F75A retained 15–25% activity towards 1NA but almost completely lost activity towards aniline (*Figure 5B*), suggesting that F75 and F86 have a greater impact on aniline than on 1NA binding in the tunnel. The mutant W235A was completely inactive, indicating the critical role of W235 for NpaA1 in proper substrate entry into the tunnel. Any combination of double mutations in NpaA1 resulted in a complete loss of activity towards both 1NA and aniline, indicating that the three aromatic amino acids act synergistically, likely participating in π-π interactions with substrates or cation-π interactions with the positively charged amino groups of aromatic amines in NpaA1. The tryptophan residues at the bottleneck of the NpaA1 and AtdA1 tunnels (W235 in NpaA1 and W81' in AtdA1, respectively) are apparently different sites that affect the tunnel's radius. We converted these residues in NpaA1 to their corresponding amino acids in AtdA1 and got the following mutations: M81A, M81W, W235L, and M81W/W235L (double mutation). The enzyme assays of the NpaA1 mutants showed that the single mutant M81A retained the ability to convert 1NA but almost completely lost activity towards aniline. However, the substitution of M81' to the corresponding tryptophan in AtdA1 caused

a reversal in substrate selectivity, from native NapA1 preferring 1NA to its mutant M81A preferring aniline. The W235L mutation in NpaA1 has a negligible effect on 1NA preference but enhances the effect of M81W in the M81W/W235L double mutant, completely eliminating 1NA conversion similar to AtdA1, with aniline becoming the only detectable substrate. (*Figure 5B*). These results suggest that the different sites of the tryptophan residues in NpaA1 and AtdA1 are crucial for influencing the differences in substrate spectra of the two enzymes. The residues M81 and W235 are the two crucial residues in NpaA1 that play a decisive role in NpaA1's ability to catalyze naphthylamine glutamylation.

## V201 is a key site influencing the selectivity of NpaA1 substrates

The substrate binding tunnel terminates at the active center. Docking analysis of NpaA1 and AtdA1 revealed V201 implicated in 1NA binding, in the NpaA1 pocket, leading to a larger naphthylamine-binding volume. In AtdA1, Y205 (conserved in GS proteins) with a larger tryptophan side-chain group may increase steric hindrance for naphthylamine's diphenyl ring (*Murray et al., 2013*; *Wang et al., 2021*). Surprisingly, the V201Y mutant in NpaA1 exhibited higher catalytic activity for both 1NA and aniline compared to the wild-type (*Figure 5B*). Single-site saturation mutagenesis was performed to investigate the effect of V201 for 1NA binding in NpaA1 (*Figure 5C*). In contrast to V201Y, V201F altered NpaA1's substrate preference, indicating the contribution of hydroxyl groups on tryptophan to enhanced substrate binding. The substitution of Val201 with Leu enhanced activity toward 1NA by 8.8-fold, indicating a more accommodating pocket for naphthylamines. Mutating V201 to hydrophobic amino acids with longer side chains increased the likelihood of catalyzing monocyclic aniline conversion. In summary, V201 is pivotal in determining substrate preference in NpaA1.

## Discussion

In contrast to well-studied microbial degradation of monocyclic aromatic amines (*Arora, 2015*), the mechanisms for microbial degradation of polycyclic aromatic amines have remained unknown until now. *Pseudomonas* sp. strain JS3066 is the first reported isolate with the ability to mineralize 1NA. The fact that enrichment from the soil at a former naphthylamine manufacturing site yielded an isolate capable of robust growth (doubling time approximately 6 hr) on 1NA suggests strongly that it plays a role in naphthylamine degradation in situ.

The mechanism involved in the degradation of 1NA by strain JS3066 was elucidated both at genetic and biochemical levels. The proposed pathway (*Figure 2D*) involves the conversion of 1NA to 1,2-dihydroxynaphthalene catalyzed by homologs of the aniline dioxygenase enzyme system. The involvement of *npaA1-npaA5* is supported strongly by both bioinformatics and experimental demonstration of the ability to catalyze the key reactions. Subsequent metabolism to salicylate is catalyzed by enzymes closely related to those of the *nag* pathway for naphthalene degradation (*Fuenmayor et al., 1998*; *Zhou et al., 2001*).

The proposed pathway for the degradation of dihydroxynaphthalene involves well-established enzymes and reactions, and is supported by the close association of the genes encoding the lower pathway for naphthalene degradation with those encoding the initial oxidation of naphthylamine. The evidence for the intermediate steps is supported by bioinformatics and analogy with established systems, but rigorous determination of the details will require additional experimental confirmation.

The strong activity of the naphthylamine dioxygenase enzyme system toward aniline as well as the high sequence similarity indicates recent divergence from a common ancestor. The fact that the isolate retains the ability to grow on aniline indicates that the initial enzyme system plays a dual role in the degradation of both aniline and naphthylamine.

Horizontal gene transfer (HGT) has played an important role in the dispersal of pathogenicity-related genes, antibiotic-resistance genes, and biodegradative genes. Bacteria respond to selective pressure exerted by anthropogenic chemicals by HGT of genetic determinants that enable the recipient to utilize the compound (*van der Meer et al., 1992*). For example, naphthalene catabolic genes (i.e. naphthalene dioxygenase) (*Herrick et al., 1997*), atrazine catabolism genes *atzABC* (*de Souza et al., 1998*), and 2,4-dichlorophenoxyacetic acid degradative genes *tfdA* *McGowan et al., 1998* have been widely disseminated in soil bacteria via HGT. Whether or not hosts acquired specific genes by HGT can be predicted by comparison of the differences in G+C content, codon usage, and phylogenies between the candidate genes and the whole genomes, as well as analyzing the presence or

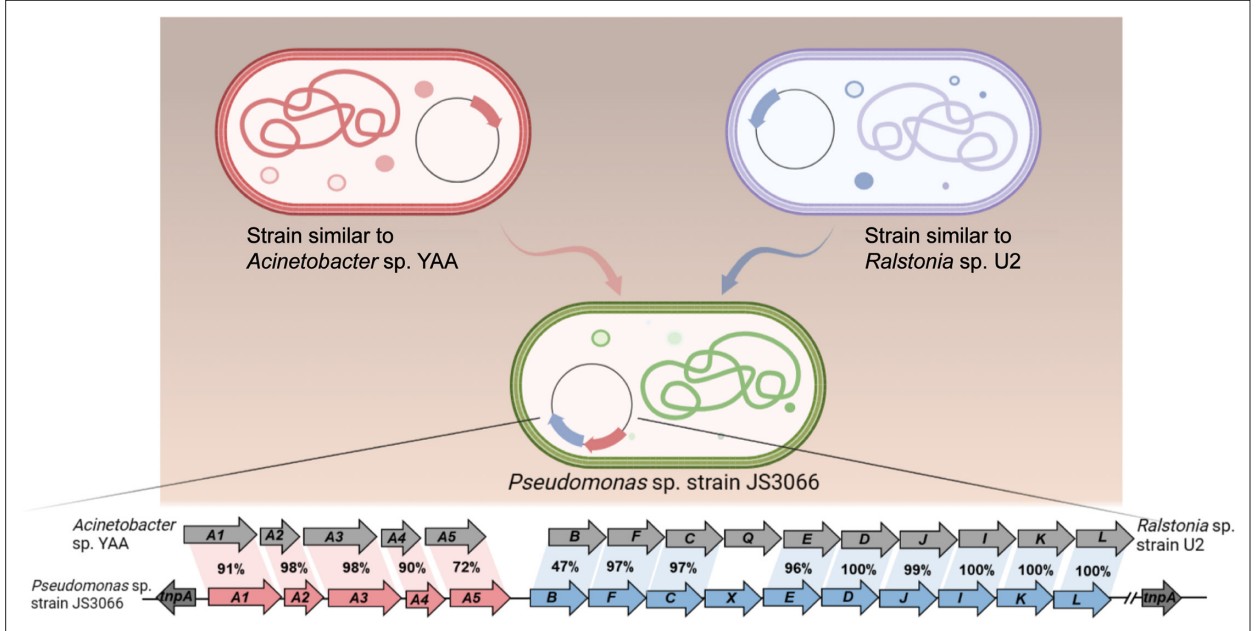

**Figure 6.** Likely sequence of horizontal gene transfer (HGT) for recruitment of genes involved in 1-Naphthylamine (1NA) degradation pathway in strain JS3066.

absence of mobile genetic elements adjacent to defined genes (**Garcia-Vallvé et al., 2000**; **Syvanen, 1994**). The G+C content of the *npaA1A2A3A4A5* gene cluster is 42.60% which is significantly lower than those of the complete genome (62.95%), *npaG* gene (67.90%), and *npaBFCXEDJIKL* genes (58.82%). The *npaA1A2A3A4A5* genes and the *npaBFCXEDJIKL* genes are flanked by transposase-encoding genes. Based on these observations, it is likely that the genes encoding enzymes that make up the 1NA degradation pathway have originated from those involved in biodegradation of aniline and of naphthalene, a process probably mediated by HGT.

During the HGT process of *npaA1A2A3A4A5*, the genes (i.e. *nagAaGHAbAcAd* in strain U2 and similar strains) encoding transformation of naphthalene into *cis*-naphthalene dihydrodiol and conversion of salicylate to gentisate within the naphthalene degradation pathway were completely replaced with the *npaA1A2A3A4A5* genes likely derived from aerobic aniline degraders (e.g. *Acinetobacter* sp. YAA and similar strains) (**Figure 6**). It is worthwhile to note that the *npaB* gene, equivalent to *nagB* in strain U2, which encodes *cis*-naphthalene dihydrodiol dehydrogenase and is located immediately downstream of the *nagAaGHAbAcAd* genes, is truncated in its 5' end probably due to substitution of the *nagAaGHAbAcAd* genes with the *npaA1A2A3A4A5* genes, therefore, serving as a tracer for assembly of 1NA degradation pathway. More importantly, the above-mentioned process led to the loss of the *nagGH* genes encoding the oxygenase component of salicylic acid 5-hydroxylase (S5H), thereby abolishing the ability of strain JS3066 to transform salicylate via gentisate even if genes (*npaJIKL*) involved in conversion of gentisate to intermediates for TCA cycle remained intact in strain JS3066. To complete the pathway of 1NA degradation, strain JS3066 employs the *npaG* gene encoding salicylic acid 1-hydroxylase (S1H) to transform salicylate into catechol prior to ring fission. The evidence supports the hypothesis that the evolution of the 1NA degradation pathway has occurred in a modular fashion, where the assembly of catabolic modules originating from different sources proceeded by transposition and subsequent rearrangement. Ancestral strains similar to *Acinetobacter* YAA and *Ralstonia* U2 were likely progenitors of JS3066, sharing common genetic traits and providing insights into its evolutionary lineage and adaptation.

The glutamylation pathway plays a crucial role in the degradation of various amine compounds (**de Azevedo Wäsch et al., 2002**; **Krysenko et al., 2019**; **Kurihara et al., 2005**; **Kurihara et al., 2008**; **Takeo et al., 2013**). When the genes encoding the aniline dioxygenase system were first discovered, one *orf* of unknown function was annotated as a 'GS-like protein' (**Rexer et al., 2006**; **de Carvalho Fernandes et al., 2022**). Subsequently, its function was established and it was renamed 'N-glutamylanilide synthase (EC6.3.1.18)' (**Takeo et al., 2013**). Several other genes in various bacteria were

similarly initially annotated as GS-like proteins and subsequently, renamed as their functions were established (*Krysenko et al., 2017*). GoaS was considered a branch of Glutamine Synthase I enzymes in the phylogeny described by *de Carvalho Fernandes et al., 2022*. However, combining the phylogenetic, function, and structure analysis in this research, the group seems sufficiently distinct from the glutamine synthase enzymes that we propose designating them 'glutamyl-organic amine synthases' (GoaS).

Previous studies indicated that the GoaS protein lost the ability to catalyze the formation of glutamine from ammonia (*de Azevedo Wäsch et al., 2002*; *Ladner et al., 2012*; *Rexer et al., 2006*; *Takeo et al., 2013*). PA5508 catalyzes the glutamylation of various aromatic amine substrates (e.g. spermidine, putrescine) (*Ladner et al., 2012*) and is the sole enzyme in GoaS with a resolved structure. Despite structural comparisons with GS, the catalytic mechanisms and binding properties of aromatic amine compounds in GoaS have not been established. In this study, we investigated the aromatic amine binding site in NpaA1 from both structural and biochemical perspectives. Despite similarities in ATP, glutamate, and manganese ion binding sites between NpaA1 and GSI, the amine binding pockets exhibit marked differences. NpaA1 evolved a large and hydrophobic substrate binding pocket to accommodate the catalysis of bulky organic amine compounds. To accommodate organic amines, both the substrate binding pocket and tunnel of NpaA1 are rich in hydrophobic amino acids. Besides hydrogen bonding interactions similar to GS proteins, hydrophobic interactions play a crucial role in facilitating substrate binding in NpaA1. GoaS lost the conserved catalytic residues found in GS during evolution, including the residues Glu357 involved in the Glu flap and the catalytic Asp 50' in *St*GS (*Figure 4—figure supplement 2*; *Gill and Eisenberg, 2001*). The conserved catalytic residues in GS are replaced by hydrophobic residues or loops in NpaA1. These differences could be significant reasons why GoaS cannot catalyze ammonium glutamylation, indicating a fundamentally different catalytic mechanism between NpaA1 and GSI.

Structural analysis indicates that different residues within both the 1NA entry tunnel and the binding pocket contribute to the broader substrate spectrum of NpaA1 compared to that of AtdA1. Specifically, the tryptophan site determines the catalytic activity towards naphthylamine. The results of mutation analysis have revealed that V201 in the substrate binding pocket of NpaA1 can only alter the preference for naphthylamine or aniline, but several mutations can enhance the activity of NpaA1, indicating a high degree of plasticity in the large substrate binding pocket of NpaA1, which is promising for future engineering endeavors on NpaA1.

The discovery of *Pseudomonas* sp. strain JS3066 and its ability to degrade 1-naphthylamine sheds light on the environmental fate of this toxic compound and provides a potential bioremediation strategy. The biochemical and structural characterization of the typical GoaS, NpaA1, for the initial reaction of 1NA biodegradation further expands our understanding of the glutamylation pathway and broadens the scope of its application in the degradation of other aromatic amines.

## Materials and methods
### Isolation and growth of 1NA-degrading bacteria
Isolates were obtained by selective enrichment under oxic conditions in a nitrogen-free minimal medium (BLK) (*Bruhn et al., 1987*) supplemented with 1-naphthylamine (0.1 mM). Subsurface samples used to inoculate the enrichments were collected from the capillary fringe at a former chemical manufacturing site in New Jersey, USA. When the 1-naphthylamine disappeared from the enrichments as determined by high-performance liquid chromatography (HPLC) the cultures were transferred to fresh medium. After the process was repeated 3 times isolates were obtained by spreading on agar plates containing BLK supplemented with 1-naphthylamine in the headspace and individual colonies were tested for the ability to grow on 1-naphthylamine in liquid medium.

### Bacterial strains, plasmids, primers, chemicals, media, and culture conditions
The bacterial strains and plasmids used in this study are described in *Supplementary file 3*, and the primers used are described in *Supplementary file 4*. *Pseudomonas* sp. strain JS3066 was grown at 30 °C in minimal medium (MM) with 0.3 mM 1NA as the sole carbon and nitrogen source. Aniline, 3,4-dichloroaniline, 2-naphthylamine, 1,5-naphthalenediamine, 1,8-naphthalenediamine,

salicylate, and catechol were purchased from Aladdin Bio-Chem Technology Co., Ltd. (Shanghai, China). 1-Naphthylamine and 1,2-dihydroxynaphthalene were supplied by Sigma-Aldrich Co., Ltd. (Shanghai, China). 3-Chloroaniline was purchased from Damas-beta Co., Ltd (Shanghai, China). 2,3-Naphthalenediamine was purchased from Meryer Chemical Technology Co., Ltd. (Shanghai, China). 2,7-Naphthalenediamine was purchased from Bide Pharmaceutical Technology Co., Ltd. (Shanghai, China). γ-Glutamylated 1NA was supplied by J&K Scientific Chemical Co., Ltd. (Shanghai, China). *Escherichia coli* strains were grown in lysogeny broth (LB) at 37 °C, whereas *P. putida* KT2440-ΔcatAΔggt was cultivated in LB at 30 °C. Kanamycin (50 μg/ml) was added to the medium as needed.

## Genome sequencing of strain JS3066 and bioinformatics

DNA of strain JS3066 was extracted with a Wizard Genomic Purification Kit (Promega, USA). The genome sequencing and assembly was done by the BGI Medical Examination Co., Ltd. (Wuhan, China) with the PacBio RSII platform. The complete genome was annotated by Rapid Annotations using the Subsystems Technology (RAST) server. The sequence of the genomic DNA is available under accession number (SUB13951314). BLASTp program was employed to deduce the amino acid identities of potential 1NA degradative genes.

To analyze phylogenetic relationships, sequences were first aligned by Clustal X version 2.1; subsequently, the phylogenetic tree was generated by the neighbor-joining method using MEGA 11 (*Tamura et al., 2021*). The evolutionary distances between branches were calculated using the Kimura two-parameter distance model, with bootstrap analysis of 1,000 resamplings to evaluate the tree topology.

## General DNA techniques

Routine isolation of genomic DNA, extraction of plasmids, restriction digestion, transformations, PCR, and electrophoresis were carried out by following standard procedures. The sequencing of PCR products and plasmids were performed by Tsingke Biotech Co., Ltd. (Shanghai, China).

## Construction of recombinant plasmids and heterologous expression

Genes from strain JS3066 were amplified using the corresponding primers *Supplementary file 4*, the resultant amplified DNA fragments were cloned into digested plasmids using a ClonExpress MultiS One Step Cloning kit (Vazyme Biotech Co., Ltd., Nanjing, China).

The vector pBBR1MCS-2 was employed for heterologous expression of suspected functional genes in *P. putida* KT2440-ΔcatAΔggt. A 5.1 kb DNA fragment containing the entire set of *npaA1A2A3A4A5* genes was amplified and then fused with the HindIII/XhoI-digested vector pBBR1MCS-2 to generate plasmid pNPA01. Afterwards, pNPA01 derivatives lacking either *npaA1*, *npaA2*, or *npaA1* and *npaA2* (*npaA1A2*) were constructed in the same way. Finally, the resulting recombinant plasmids harboring different gene combinations (namely *npaA1A2A3A4A5*, *npaA1A3A4A5*, *npaA2A3A4A5*, and *npaA3A4A5*) were introduced into *P. putida* KT2440-ΔcatAΔggt by electroporation to yield the recombinants KT/pNPA01, KT/pNPA01-ΔA1, KT/pNPA01-ΔA2, and KT/pNPA01-ΔA12, respectively.

For overexpression of the *npaA1* gene, it was amplified from the genomic DNA of strain JS3066 by PCR and then cloned into pET-29a to obtain the expression construct pET-*npaA1* which was transformed into *E. coli* BL21 (DE3). The expression and purification of NpaA1, an N-terminal Strep II-tagged fusion protein, were performed according to procedures described previously (*Ji et al., 2019*). The eluted proteins were further fractionated by gel filtration on a Superdex 200 Increase 10/300 GL column (Cytiva) with the buffer containing 50 mM Tris-HCl and 200 mM NaCl. The purity of Strep II-tagged NpaA1 was analyzed by 12.5% SDS-PAGE. Protein concentration was determined by using the Bradford method. The expression and purification of NpaG and AtdA1, also N-terminal Strep II-tagged fusion proteins, were performed in the same way as for NpaA1. The primers used for constructing mutant vectors are listed in *Supplementary file 5*, and the mutants of NpaA1 were overexpressed and purified by the same methods as above.

## SEC-Multi angle light scattering (SEC-MALS)

SEC-MALS was used to determine the molecular weight of NpaA1. Purified NpaA1, separated by gel filtration, was diluted to a final concentration of 2 mg/ml and dissolved in a 50 mM Tris buffer for sample loading.

## Crystallization and data collection

Purified NpaA1 protein was concentrated at 12 mg/ml in the buffer containing 30 mM Tris-HCl (pH 8.0) and 120 mM NaCl. Crystals were obtained at 20 °C in 1–2 weeks by sitting-drop vapor diffusion. Apo-NpaA1 was obtained in the buffer containing 0.1 M magnesium chloride hexahydrate, 0.1 M sodium acetate trihydrate, 0.1 M Bis-Tris 6.5 and 15 % v/v PEG smear broad (the ratio of protein to reservoir solution was 1:2). To obtain crystals of NpaA1–ADP–MetSox-P complex, ATP and MetSox were added to NpaA1 to a final concentration of 2 mM, and the protein solution was mixed 1:2 with the buffer containing 0.1 M magnesium chloride hexahydrate, 0.1 M rubidium chloride, 0.1 M PIPES 7.0 and 20 % v/v PEG smear low. The NpaA1–ADP was obtained under the same conditions as the NpaA1–ADP–MetSox-P complex above.

All the X-ray diffraction data were collected on the BL19U1 beamline at the Shanghai Synchrotron Radiation Facility. The initial data were processed by the HKL3000 program.

## Structure determination and refinement

The crystal structure of apo-NpaA1 was determined by molecular replacement using the model of NpaA1 built by Alphafold2 (*Jumper et al., 2021*). The structure of apo-NpaA1 was used as the model for the other structures. The refinements of these structures were performed using Coot (*Emsley et al., 2010*) and Phenix (*Liebschner et al., 2019*).

## Similarity searches and sequence comparison

Amino acids of NpaA1 were used to search for similar sequences in the Swissprot database. Sequences with similarity above 27% were selected for multiple sequence alignment. Multiple sequence alignment and phylogenetic tree construction were performed using MEGA 11 (*Tamura et al., 2021*).

## Biotransformation of 1NA in cell suspensions of *P. putida* KT2440-Δ*catA*Δ*ggt* harboring various recombinant plasmids

Recombinants KT/pNPA01, KT/pNPA01-Δ*A1*, KT/pNPA01-Δ*A2*, and KT/pNPA01-Δ*A12* were individually grown in 250 ml Erlenmeyer flasks with 100 ml of LB medium containing kanamycin (50 µg/ml) at 30 °C and 180 rpm, harvested by centrifugation (4 °C, 6000 rpm, 5 min), washed twice with Tris-HCl buffer (50 mM, pH 8.0) and finally resuspended in the same buffer. The optical density at 600 nm ($OD_{600}$) of cell suspensions was adjusted to approximately 8.0. The substrate 1NA (final concentration of 0.4 mM) was added to the suspension, and degradation experiments were performed at 30 °C and 170 rpm on a rotary shaker. Samples were collected at regular intervals, and the change in concentrations of 1NA was analyzed by HPLC.

## Enzyme assays

Activities of NpaA1 and AtdA1 against different substrates were analyzed spectrophotometrically with a Lambda 25 spectrophotometer (PerkinElmer/Cetus, Norwalk, CT) by following the disappearance of tested substrates at individually defined wavelengths. The reaction system (0.5 ml volume) contained 1.5 mM ATP, 1.0 mM L-glutamate, 2.0 mM $MgCl_2$, and 0.28 mg of NpaA1 in 50 mM Tris-HCl buffer (pH 8.0). The assay was initiated by adding different substrates. The molar extinction coefficients of various potential substrates were obtained by measuring their absorbance values in the reaction buffer at each characteristic wavelength. As for NpaA1 and AtdA1, one unit of enzyme activity (U) is defined as the amount of enzyme required for the consumption of 1 µmol of substrate in 1 min at 30 °C. Specific activities for NpaA1 and AtdA1 are expressed as units per gram of protein. The activity of NpaG against salicylate was measured by monitoring NADH oxidation at 340 nm and the molar extinction coefficient for NADH was taken as 6220 $M^{-1} \cdot cm^{-1}$. The reaction system (0.5 ml volume) contained 200 µM NADH, 40 µM FAD, 80 µM salicylate, and 0.38 µg of NpaG in 50 mM potassium phosphate buffer (pH 7.2). For NpaG, one unit of enzyme (U) is defined as the amount of enzyme required for the consumption of 1 µmol of NADH in 1 min at 30 °C. Specific activity for NpaG against salicylate is expressed as units per milligram of protein.

## Analytical methods

To isolate and identify the metabolites, the biotransformation sample was acidified to pH 2 with concentrated HCl and then extracted with an equal volume of ethyl acetate which was subsequently

removed by evaporation. Bis (trimethylsilyl) trifluoroacetamide (BSTFA) and trimethylchlorosilane (TMCS) (volume ratio: 99/1) was used as the derivatization reagent so that active hydrogen atom(s) of the 1NA metabolites were replaced by a trimethylsilyl (TMS) group (Si(CH$_3$)$_3$), m/z 73. The pellet was dissolved in BSTFA-TMCS and then incubated at 60 °C for 30 min prior to GC-MS analysis. GC-MS analyses were performed on a TRACE 1310 gas chromatograph (Thermo Fisher Scientific, MA, USA) using a capillary column HP-5MS (0.25 mm × 30 m, Agilent Technologies, CA, USA). The column temperature gradient was 0–5 min, 60 °C; 5–27 min, 60–280°C (10 °C min$^{-1}$); 27–32 min, 280 °C. The detector was a TSQ8000 Triple Quadrupole MS (Thermo Fisher Scientific, USA). The following conditions were applied for mass analysis: ionization mode, EI$^+$; ionizing electron energy, 70 eV; source temperature, 250 °C, and mass range m/z 0–500. Mass spectra of individual total ion peaks were identified by comparison with authentic standards and the *Wiley*.275L mass spectra data base.

HPLC analysis of substrates and their oxidation products was performed on a Waters e2695 Separation Module equipped with a C18 reversed-phase column (5 μm, 4.6 × 250 mm).The mobile phase consisted of water containing 0.1% (vol/vol) acetic acid (A) and methanol (B) at a flow rate of 1.0 ml min$^{-1}$. A gradient elution program was as follows: 0–15 min 20–80% B; 15–25 min 80% B; 25–25.1 min 80–20% B; 25.1–40 min 20% B. The column temperature was 30 °C and the injection volume was 20 μl. Qualitative analysis of the oxidation product from the reaction of 1NA degradation catalyzed by NpaA1 was executed with ultra-performance liquid chromatography-quadrupole-time of flight-mass spectrometry (UPLC-QTOF MS), with the electrospray ionization (ESI) source in positive ion mode.

## Acknowledgements

We thank the team of beamline BL18U1 in the Shanghai Synchrotron Radiation Facility for diffraction data collection, Wei Zhang for assistance with LC-MS experiments, Jianting Zheng for crystallization experiments, and members of the Ning-Yi Zhou's lab for helpful discussions. Funding was provided by National Key Research and Development Program of China 2021YFA0909500 (NYZ) and Chemours Corporate Remediation Group (JCS). The funders had no role in study design, data collection and interpretation, or the decision to submit the work for publication.

## Additional information

### Funding

| Funder | Grant reference number | Author |
|---|---|---|
| National Key Research and Development Program of China | 2021YFA0909500 | Ning-Yi Zhou |
| Chemours Corporate Remediation Group | | Jim C Spain |

The funders had no role in study design, data collection and interpretation, or the decision to submit the work for publication.

### Author contributions

Shu-Ting Zhang, Conceptualization, Validation, Investigation, Methodology, Writing – original draft, Writing – review and editing; Shi-Kai Deng, Conceptualization, Validation, Investigation, Methodology, Writing – original draft; Tao Li, Conceptualization, Methodology, Writing – review and editing; Megan E Maloney, Validation, Investigation, Methodology; De-Feng Li, Software, Validation, Methodology; Jim C Spain, Conceptualization, Supervision, Funding acquisition, Investigation, Writing – review and editing; Ning-Yi Zhou, Conceptualization, Supervision, Funding acquisition, Project administration, Writing – review and editing

### Author ORCIDs

Shu-Ting Zhang ⓘ https://orcid.org/0000-0003-0783-4529
Tao Li ⓘ https://orcid.org/0000-0002-8255-7798
De-Feng Li ⓘ https://orcid.org/0000-0002-8683-019X

Jim C Spain 🔗 http://orcid.org/0000-0003-1779-1972
Ning-Yi Zhou 🔗 https://orcid.org/0000-0002-0917-5750

**Decision letter and Author response**
Decision letter https://doi.org/10.7554/eLife.95555.sa1
Author response https://doi.org/10.7554/eLife.95555.sa2

## Additional files

**Supplementary files**
• Supplementary file 1. Identities of ORFs from gene clusters responsible for 1-Naphthylamine (1NA) degradation.
• Supplementary file 2. Crystallographic data collection and refinement of NpaA1 and its complex.
• Supplementary file 3. Strains and plasmids used in this study.
• Supplementary file 4. Primers used in this study.
• Supplementary file 5. Primers used for constructing mutant vectors.
• MDAR checklist

**Data availability**
The complete genome of Pseudomonas sp. strain JS3066 is available in the NCBI database under BioProject identifier (ID) PRJNA1035437 or BioSample accession SUB13951314. The structures of apo-NpaA1, NpaA1-AMPPNP, NpaA1-ADP-MetSox-P, have been deposited in PDB (https://www.rcsb.org/) under accession codes 8X6Z, 8WWU, 8WWV, respectively. All data generated or analyzed during this study are included in the manuscript and supplementary files.

The following datasets were generated:

| Author(s) | Year | Dataset title | Dataset URL | Database and Identifier |
|---|---|---|---|---|
| Zhang ST | 2023 | An aromatic amine degrader, strain Pseudomonas sp. strain JS3066 | https://www.ncbi.nlm.nih.gov/bioproject/PRJNA1035437 | NCBI BioSample, PRJNA1035437 |
| Zhang ST | 2023 | 1-naphthylamine GS from Pseudomonas sp. JS3066 | https://www.rcsb.org/structure/8X6Z | RCSB Protein Data Bank, JS3066 |
| Zhang ST | 2023 | 1-naphthylamine GS in complex with AMP PNP | https://www.rcsb.org/structure/8WWU | RCSB Protein Data Bank, 8WWU |
| Zhang ST | 2023 | 1-naphthylamine GS in complex with ADP and MetSox-P | https://www.rcsb.org/structure/8WWV | RCSB Protein Data Bank, 8WWV |

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
