## [Editor Report]

This important work identifies a *P. aeruginosa* strain and enzyme that can degrade 1-naphthylamine, a harmful industrial pollutant. Data resulting from in vivo, biochemical, and structural approaches are compelling. This paper would be of major interest to biologists and enzymologists studying biodegradation of industrial pollutants.

---

## [Decision Letter]

**Decision letter after peer review:**

Thank you for submitting your article "Discovery of the 1-naphthylamine biodegradation pathway reveals an enzyme that catalyzes 1-naphthylamine glutamylation" for consideration by *eLife*. Your article has been reviewed by 2 peer reviewers, and the evaluation has been overseen by a Reviewing Editor and Qiang Cui as the Senior Editor.

Essential revisions (for the authors):

1) Although the study is well designed with various strong biochemical and cellular data, we felt that additional mutagenesis studies are necessary to bolster the main claim and support the broad substrate specificity.

*Reviewer #2 (Recommendations for the authors):*

Comments:

1. The title and abstract may be revised to reflect the whole manuscript or the key aspects. Currently, they do not give a clear impression on the whole story or the key findings.

2. γ-Glutamylorganoamide synthetase and naphthylene dioxygenase are two enzymes, as they are not forming a complex nor catalyze a single reaction.

---

## [Author Response]

Essential revisions (for the authors):Reviewer #2 (Recommendations for the authors):Comments:1. The title and abstract may be revised to reflect the whole manuscript or the key aspects. Currently, they do not give a clear impression on the whole story or the key findings.

Thank you very much for your suggestion. Our work primarily consists of elucidating the biodegradation pathway of 1-naphthylamine and investigating the substrate selectivity mechanism of the key enzyme NpaA1 in this pathway. Due to space limitations, the original title may not have fully conveyed the whole story. To emphasize both aspects, we have revised the title to: “Discovery of the 1-naphthylamine biodegradation pathway reveals a broad-substrate-spectrum enzyme catalyzing 1-naphthylamine glutamylation”.

I have also made modifications to the abstract, and the revised version is as follows: “1-Naphthylamine (1NA), which is harmful to human and aquatic animals, has been used widely in the manufacturing of dyes, pesticides, and rubber antioxidants. Nevertheless, little is known about its environmental behavior and no bacteria have been reported to use it as the growth substrate. Herein, we describe a pathway for 1NA degradation in the isolate *Pseudomonas* sp. strain JS3066, determine the structure and mechanism of the enzyme NpaA1 that catalyzes the initial reaction, and reveal how the pathway evolved. From genetic and enzymatic analysis, a five gene-cluster encoding a dioxygenase system was determined to be responsible for the initial steps in 1NA degradation through glutamylation of 1NA. The g-glutamylated 1NA was subsequently oxidized to 1,2-dihydroxynaphthalene which was further degraded by the well-established pathway of naphthalene degradation via catechol. A glutamine synthetase-like (GS-like) enzyme (NpaA1) initiates 1NA glutamylation, and this enzyme exhibits a broad substrate selectivity toward a variety of anilines and naphthylamine derivatives. Structural analysis revealed that the aromatic residues in the 1NA entry tunnel and the V201 site in the large substrate-binding pocket significantly influence NpaA1's substrate preferences. The findings enhance understanding of degrading polycyclic aromatic amines, and will also enable the application of bioremediation at naphthylamine contaminated sites.”.

2. γ-Glutamylorganoamide synthetase and naphthylene dioxygenase are two enzymes, as they are not forming a complex nor catalyze a single reaction.

Yes, γ-glutamylorganoamide synthetase and naphthalene dioxygenase are two enzymes that catalyze distinct reactions. They are the components of naphthylamine dioxygenase system. The deamination of aniline involves two steps: glutamylation and dioxygenation. Since the genes encoding enzymes catalyzing both steps are often clustered together, studies have referred to aniline glutamine synthetase and glutamine aniline dioxygenase as constituting the aniline dioxygenase system (AD system)^[1]^, a terminology widely accepted in the field^[2, 3]^. This convention also applies to the naphthylamine dioxygenase system discussed in this study.